# 3D sub-diffraction imaging in a conventional confocal configuration by exploiting super-linear emitters

Denitza Denkova [1,5,8], Martin Ploschner [1,6,8], Minakshi Das [2], Lindsay M. Parker [2], Xianlin Zheng[1], Yiqing Lu [1], Antony Orth[3,7], Nicolle H. Packer [2,4] & James A. Piper [1]

Sub-diffraction microscopy enables bio-imaging with unprecedented clarity. However, most super-resolution methods require complex, costly purpose-built systems, involve image post-processing and struggle with sub-diffraction imaging in 3D. Here, we realize a conceptually different super-resolution approach which circumvents these limitations and enables 3D sub-diffraction imaging on conventional confocal microscopes. We refer to it as super-linear excitation-emission (SEE) microscopy, as it relies on markers with super-linear dependence of the emission on the excitation power. Super-linear markers proposed here are upconversion nanoparticles of $NaYF_4$, doped with 20% Yb and unconventionally high 8% Tm, which are conveniently excited in the near-infrared biological window. We develop a computational framework calculating the 3D resolution for any viable scanning beam shape and excitation-emission probe profile. Imaging of colominic acid-coated upconversion nanoparticles endocytosed by neuronal cells, at resolutions twice better than the diffraction limit both in lateral and axial directions, illustrates the applicability of SEE microscopy for subcellular biology.

[1] ARC Centre of Excellence for Nanoscale BioPhotonics (CNBP), Department of Physics and Astronomy, Macquarie University, Sydney, NSW 2109, Australia. [2] ARC Centre of Excellence for Nanoscale BioPhotonics (CNBP), Department of Molecular Sciences, Macquarie University, Sydney, NSW 2109, Australia. [3] ARC Centre of Excellence for Nanoscale BioPhotonics (CNBP), School of Science, RMIT University, Melbourne, VIC 3000, Australia. [4] ARC Centre of Excellence for Nanoscale BioPhotonics (CNBP), Institute for Glycomics, Griffith University, Southport, QLD 4215, Australia. [5] Present address: Bioengineering in Reproductive Health Group, Institute for BioEngineering of Catalonia (IBEC), 08028 Barcelona, Spain. [6] Present address: School of Information Technology and Electrical Engineering, The University of Queensland, Brisbane, QLD 4072, Australia. [7] Present address: National Research Council of Canada, Ottawa, Ontario K1K 3Y2, Canada. [8] These authors contributed equally: Denitza Denkova, Martin Ploschner. Correspondence and requests for materials should be addressed to D.D. (email: denitza.denkova@gmail.com) or to M.P. (email: ploschner.m@gmail.com)

The resolution of conventional lens-based microscopes is limited by diffraction to about $\lambda/2$ in lateral (in-plane) and $\lambda$ in the axial (out-of-plane) direction[1]. The growing demand from the biological sciences to optically image nanoscale features has led to the development of numerous techniques, which push the resolving power of optical microscopes beyond this diffraction barrier. These super-resolution methods have helped answer biological questions by visualization of nano-sized structures and interactions which were otherwise not experimentally resolvable[2–4].

Super-resolution techniques are typically based on one of the two approaches: localization of individual molecules (photo-activated localization (PALM)[5], stochastic optical reconstruction (STORM)[6], single-molecule active control (SMACM) microscopy[7]) or on engineering of the illumination and detection pathways (stimulated emission depletion (STED)[8], structured illumination (SIM)[9], 4Pi[10], image interference ($I^nM$)[11], MIN-FLUX microscopy[12]). Each technique has its own set of trade-offs, but typically suffers from one or more of the following drawbacks: (i) requires purpose-built optical system and software, usually with high cost and complexity (STED, PALM, STORM, SIM); (ii) necessitates acquisition of many images and their subsequent post-processing, which is time consuming and prone to imaging artifacts (PALM, STORM, SIM); (iii) needs high laser fluence, often at visible wavelengths, which is potentially damaging to biological samples, results in rapid bleaching of the fluorophores and obstructs long-time imaging and tracking[13]; (iv) lacks biomarkers working in the less photo-toxic, biologically convenient near-infrared window[14,15]. Moreover, 3D optical sectioning and tracking applications typically require sub-diffraction resolution simultaneously both in lateral and axial directions. This is supported by only a handful of super-resolution techniques[16,17], often with a significant penalty in terms of photon budget. The goal remains for 3D super-resolution techniques to overcome these practical hindrances, and to become established as everyday tools in biology labs.

At the end of last century, a conceptually different super-resolution approach had been suggested, having the potential to circumvent the above limitations and conveniently achieve 3D sub-diffraction imaging on a standard confocal microscope, without the need for setup modifications or image processing[18,19]. The prerequisite for the method to work is to use an unconventional class of luminescent markers, so called super-linear emitters, for which the emission depends in a super-linear fashion on the excitation power. In brief, whenever a laser beam scans over such a marker, it is only the most intense, central part of the beam that yields significant emission. As this occurs in a region smaller than the size of the beam itself, the imaging resolution is effectively improved. The stronger the super-linearity, the smaller the region of significant emission, and therefore the better the resolution.

So far, this simple idea has found limited practical realization, mainly due to the lack of suitable fluorophores. Luminescent molecules with sufficiently strong super-linear properties are restricted to visible wavelengths and turned out difficult to design, synthesize and apply in practice[18–23]. From the library of luminescent nanoparticles, nanodiamonds work only in the visible spectrum, and require complex conditions to enter the super-linear regime of operation, involving several different lasers and specific illumination sequence[24]. This has so far prevented the use of super-linear nanodiamonds in a biological setting. Quantum dots were reported as a promising prospect in live cells, demonstrating a potential for sub-diffraction imaging due to super-linearity[25]. However, the studied quantum dots work only in the visible range and require a tri-exciton process, which is excited at 2 orders of magnitude higher power than the standard diffraction-limited mono-exciton emission.

In contrast to the labels discussed above, upconversion nanoparticles (UCNPs) can enter into the super-linear regime spontaneously, without the need for complex procedures or higher excitation powers. UCNPs are already extensively exploited as biological probes, mainly due to their high photostability, tunable narrow emission lines and long lifetimes, promising for multiplexing applications[26,27]. Moreover, UCNPs are typically excited in the near-infrared window, which allows deeper penetration in biological samples and improves the signal-to-noise ratio by suppressing native autofluorescence[28]. Several reports have shown that UCNPs can be imaged with sub-diffraction resolution on a confocal microscope[29–31]. However, these studies are constrained to an experimental demonstration of lateral resolution improvement, in a proof-of-principle setting, and do not present a comprehensive theory with predictive power for a realistic confocal system and probe emission profile. This has limited the uptake of the method and demonstration in a biological specimen has not yet been realized.

Overall, despite the revived attention in recent years, the following key challenges remain to be tackled in order to establish this promising idea as a routine capability in biological labs: (i) availability of convenient super-linear fluorophores, especially in the near-infrared range; (ii) development of a theoretical framework, capable to calculate both the lateral and the axial resolution at practical experimental conditions; (iii) demonstration of the technique in a biological environment.

Here, we achieve these milestones and for the first time, we provide strong experimental evidence supported by rigorous simulations that the super-linear properties of UCNPs can be used to achieve 3D super-resolution imaging in a biological specimen in a simple confocal setup. We develop a universal theoretical and computational framework, which can be used to calculate the enhancement of the optical resolution for both axial and lateral directions when imaging arbitrary super-linear fluorophores under an arbitrary excitation beam. We apply the developed theory to investigate the advantages that our particular choice of super-linear emitters (UCNPs of $NaYF_4$: 20% Yb, 8% Tm) can offer in terms of sub-diffraction imaging in a confocal microscope. These particles are accessible and convenient for biological applications, because they are based on a commonly used type of UCNPs and are excited in the near-infrared region. In contrast to all other super-resolution techniques, we demonstrate improvement of the resolution in the sub-diffraction regime by lowering the photon budget—a counter-intuitive trend which paves the way toward low-power super-resolution imaging in biological applications. Finally, we apply the technique in a biological setting by imaging colominic acid functionalized UCNPs, uptaken by neuronal cells. We exceed the diffraction limit by a factor of two for both lateral and axial directions. To avoid terminological confusion, we refer to the methodology developed in this paper as superlinear excitation–emission (SEE) microscopy (see Discussion section). If the imaging is realized with upconversion nanoparticles, we will refer to it as upconversion super-linear excitation–emission (uSEE) microscopy.

## Results

**Experimental proof-of-principle.** We achieve sub-diffraction imaging via SEE microscopy simply by scanning a super-linear UCNP in a conventional confocal configuration (Fig. 1a).

As super-linear markers, we employ $NaYF_4$ upconversion nanoparticles ($\beta$-phase; 46 nm average size), doped with 20% Yb (sensitizer) and 8% Tm (activator), which are fixed in an index-matching media between a coverslip and a microscope slide (see Methods section and Supplementary Methods). Briefly, upconversion is an optical process in which lower-energy photons are

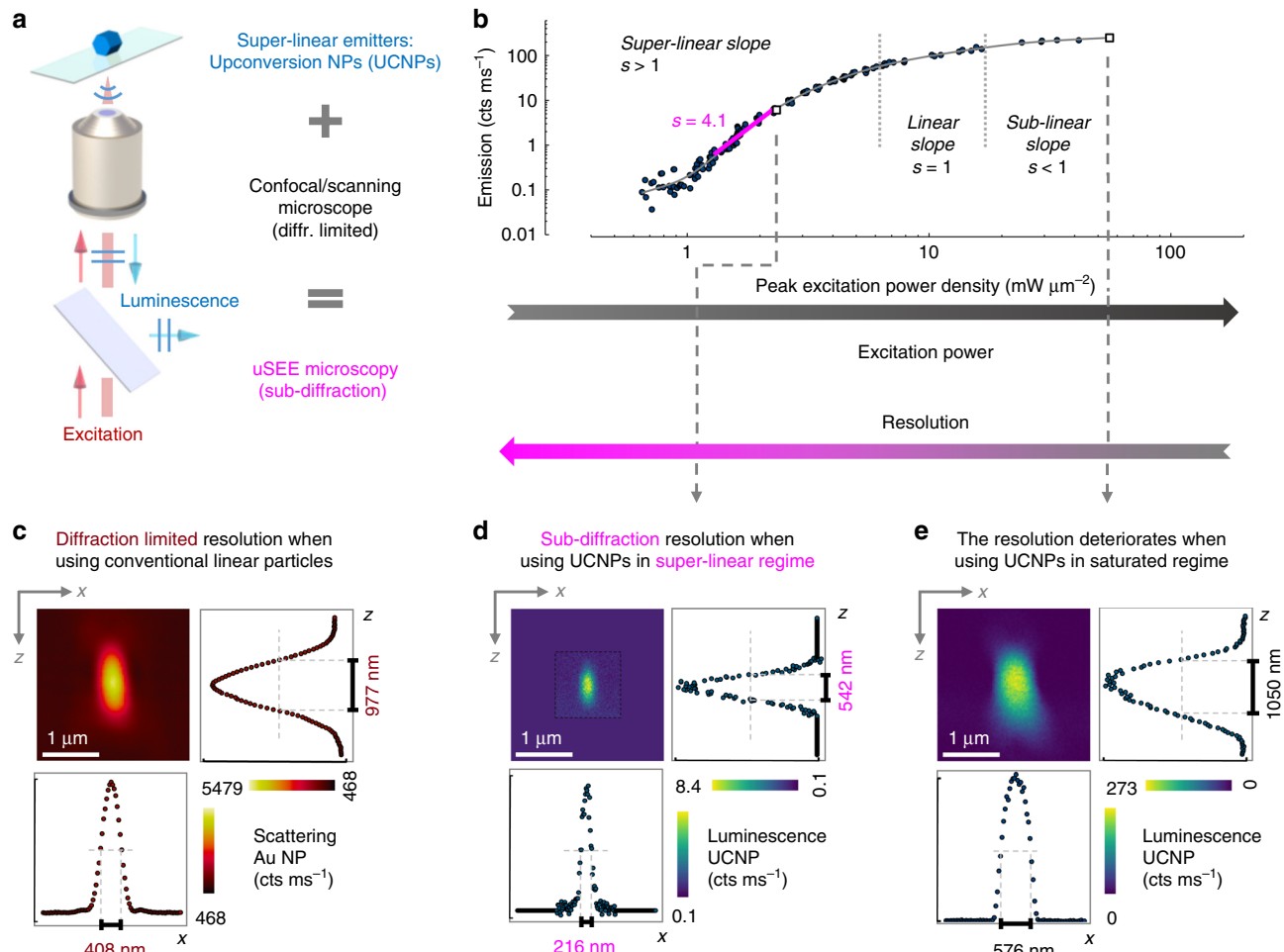

**Fig. 1** 3D sub-diffraction resolution achieved by SEE microscopy. **a** Sub-diffraction resolution can be achieved in a conventional confocal microscope when using super-linear emitters, in our case upconversion nanoparticles (UCNPs of NaYF$_4$: 20% Yb, 8% Tm). **b** The emission from the UCNPs at $\lambda = 455$ nm, exhibits a strong super-linear dependence on the excitation power, with a maximum of slope $s = 4.1$ in the low-power range. The open square symbols represent the peak power densities at which the measurements in panel (**d-e**) were obtained. **c** Experimentally, the diffraction limit of the setup is determined from the PSF of the excitation beam, obtained by scanning a gold particle through the beam and recording the scattering intensity at each point. The diffraction-limited lateral ($x$ direction) resolution is 408 nm and the axial ($z$ direction) resolution is 977 nm as shown in the corresponding cross sections. In contrast, when using non-linear emitters (**d-e**), the size of the PSF, representing the resolution, changes depending on the excitation power. **d** In the low-power super-linear range we achieve sub-diffraction imaging (uSEE microscopy), with lateral resolution 216 nm ($\lambda/4.5$ compared with the diffraction limit of $\lambda/2.4$) and axial resolution 542 nm ($\lambda/1.8$ compared with the diffraction limit of $\lambda$). **e** In the high-power saturation regime the resolution is worse than the diffraction limit. The dashed black box in (**d**) indicates the span of the experimentally measured area. The area outside of the black box contains a flat background equal to the average background in the measured area. This facilitates the visual comparison of panels (**c-e**)

converted to higher-energy photons. In our case, the Yb$^{3+}$ ions act as a sensitizer, absorbing the infrared radiation of the 976 nm excitation laser. Then, via non-radiative energy transfer, they excite a series of levels in the activator Tm$^{3+}$ ions in a step-wise manner. In addition, various energy re-distribution processes can take place, for example, cross relaxation between neighboring Tm$^{3+}$ ions, excited state absorption and photon avalanche[32–34]. Finally, the Tm$^{3+}$ ions emit at various visible and ultra-violet wavelengths. The number of photons involved in the above-described processes depends on the particular UCNP composition and the excitation intensity. In the context of uSEE microscopy, it is important to note that although the emissions have various distinct wavelengths, they are all excited at the same 976 nm wavelength.

It has been shown that upconversion processes can yield non-linear, and in particular super-linear emission[34,35]. The emission depends on the excitation intensity to the power of $s$, where $s$ is a number equal or smaller than the number of photons $N$ involved

in the process[36]. The process is considered linear for $s = 1$, super-linear for $s > 1$ and sub-linear for $s < 1$. As a rule of thumb, the bigger the value of $s$, the stronger the super-linearity and, respectively, the more suitable the upconversion emission is for uSEE microscopy.

The transition of particular interest for this paper is the emission from level $^1D_2$ to $^3F_4$ of the Tm$^{3+}$ ions with emission peak at 455 nm. This transition has been referred to as $N = 4, 5, 6$ photon process or a mixture of these[33,37,38]. Due to the high number of photons involved, the 455 nm emission line has the potential to yield relatively strong non-linearity, compared with other upconversion processes reported in the literature with typical values of $N = 2$ or $3$[32,39].

A typical experimental excitation–emission curve, obtained from an isolated UCNP, illustrates the dependence of the 455 nm emission on the excitation power density (Fig. 1b). In a log–log plot, the slope (i.e., gradient) of the curve at a certain excitation power corresponds to the value of $s$ at that power. At

low excitation power density, when the peak value is below $1 \, mW \, \mu m^{-2}$, the particle is barely luminescent. Further increase of the excitation power, until about peak density of $2.5 \, mW \, \mu m^{-2}$, results in steep increase of the luminescence intensity. In this range, the particle is in the super-linear regime, and the slope of the graph is $s > 1$. The highest slope we achieve for this particle is $s = 4.1$. When the excitation power is increased further, the particle goes through a linear ($s = 1$), then sub-linear ($s < 1$) regime and finally reaches a saturation level with emission of about $250 \, cts \, ms^{-1}$. In this paper, when we use the term super-linear emitter, we are referring to a non-linear emitter in its super-linear regime.

The confocal microscope in which we demonstrate uSEE microscopy uses a 976 nm laser excitation beam, circularly polarized and focused through a 1.4 NA, ×100 objective. The luminescence emission is collected through the same objective, separated from the laser line by a dichroic mirror and directed to an avalanche photodiode (APD) through a narrow-band filter to isolate the 455 nm emission line of the $Tm^{3+}$ ions. The APD is synchronized with the XYZ scanning piezo stage to generate a luminescence image. Further details about the experimental setup can be found in Supplementary Methods and Supplementary Fig. 6.

To experimentally determine the diffraction-limited resolution of our setup, we refer to the commonly used Rayleigh criteria[40]. According to this criteria, a reasonable estimate for the resolution is the full-width-at-half-maximum (FWHM) of the point spread function (PSF), where the PSF is the minimum sized feature that can be formed by the optical system. We experimentally obtain the PSF, by using a well-established method of scanning a gold nanoparticle (80 nm in size) over the 976 nm excitation beam (Fig. 1c and Supplementary Methods). The lateral and axial cross sections in Fig. 1c allow us to measure the FWHM, and respectively, to experimentally determine the diffraction-limited resolution of our system: 408 nm in lateral direction and 977 nm in axial direction. These values are in close agreement with the theoretically expected values.

A confocally scanned nanoparticle experiences a wide range of excitation power densities, spanning from zero to the peak intensity of the beam. In the case of a gold nanoparticle, a fraction of this excitation light scatters to the detector. Crucially, the scattered fraction is independent of the excitation power. In such a scenario, equivalent to a linear excitation–emission curve behavior, the shape and size of the PSF shown in Fig. 1c is representative for both the excitation beam and the scattering profile of the particle. As the excitation beam is diffraction-limited, the scattering profile is also diffraction-limited. At different excitation powers, the scattered intensity rescales, but the FWHM, i.e., the resolution, stays diffraction limited. This conclusion is generally valid for conventional confocal microscopy, employing any linear scatterer/emitter. The situation is entirely different when we use a non-linear emitter, such as the UCNP with excitation–emission curve shown in Fig. 1b. Due to the non-linear character of the excitation–emission curve, the emission profile can have a different shape than the excitation beam profile. Thus, the FWHM and, respectively, the resolution can vary with the excitation power as demonstrated in Fig. 1d, e.

When the peak power of the excitation beam is within the steep super-linear range of the excitation–emission curve ($\approx$1.3–2.5 mW $\mu m^{-2}$), the confocal microscope is spontaneously operating in uSEE microscopy mode. The wings of the beam barely excite the particle and only the central, most intense part of the beam yields a strong emission from the UCNP. As a result, the emission profile has a narrower and steeper shape than the excitation profile. The FWHM effectively shrinks and at a peak excitation power density of 2.3 mW $\mu m^{-2}$ the resolution is twice

better than the diffraction limit, both in lateral 216 nm ($\lambda$/4.5) and axial 542 nm ($\lambda$/1.8) directions (Fig. 1d). To facilitate visual comparison between images with different scanning range, a flat background, equal to the average image background, has been added around the experimental image (the area outside of the dashed black box in Fig. 1d). A similar visual aid is used in Figs. 2 and 3.

At higher excitation powers, the excitation–emission curve starts to flatten. Both the wings and the central part of the beam have sufficient power to draw close to maximum emission from the particle. As a result, the emission profile broadens, and the resolution at such high powers is worse than the diffraction limit of the setup. An example of this case is presented in Fig. 1e, where the resolution is 576 nm in lateral and 1050 nm in axial direction for imaging at peak power density of 55.8 mW $\mu m^{-2}$.

The nanoparticles show high photostability under uSEE microscopy imaging conditions and can be imaged for hours without significant deterioration of the emission intensity and without change in the uSEE microscopy resolution (Supplementary Note 1). However, under high-power illumination for prolonged time periods, necessary in this paper for the verification of the theoretical framework across a broad excitation range, the emission from the particle starts to deteriorate and the excitation–emission curve changes. To avoid such effects, we have applied a pre-illumination procedure to the particle used in Figs. 1 and 3, which leads to reduced, but stable emission (Supplementary Methods). The rest of the data in the paper are obtained under normal imaging conditions, where such procedure is not needed.

Ultimately, from the end-user point of view, the parameter which matters in terms of resolution, is the minimum distance at which neighboring particles can be resolved. The most commonly used Rayleigh criteria states that two particles are resolvable if the intensity dip between the particles is at least 26.3% of the emission peak intensity[40]. In the ideal situation considered by Rayleigh, the background noise is neglected and the minimum resolvable distance between the particles equals the FWHM of the particle's PSF. Typically, it is preferred to use the FWHM value to determine the resolution, as it is easy to measure experimentally.

However, any practical experiment involves a certain background/detector noise level. If the PSF wings drop significantly below the noise level, the PSF maximum and the FWHM values can be underestimated and particles separated by the FWHM distance will not be resolvable. Thus, in order to justify the use of FWHM as a resolution criteria in our paper, we need to verify that in our particular experiments the minimum distance between resolvable particles equals the FWHM.

We image two particles, separated by ca. 200 nm (Fig. 2). Illuminating the particles with a beam of 14.5 mW $\mu m^{-2}$ peak excitation power density generates a diffraction-limited image in which the particles are not resolvable, as a single emission peak is observed in the confocal image and the corresponding cross-section (Fig. 2a).

Reducing the peak power density to 1.2 mW $\mu m^{-2}$ spontaneously switches the confocal microscope to uSEE mode (Fig. 2b). As the dip between the emission peaks is close to half of the emission peak values, the particles are clearly resolvable according to Rayleigh's criteria. The peak-to-peak distance of 206 nm between the particles is almost identical to the FWHM of a single particle emission profile (202 nm) at this power. Thus, we conclude that for the typical power range used for uSEE microscopy in our work, the FWHM of a single particle emission profile is a good estimator of the system's resolution.

In the axial direction, we were unable to find particles spaced at a suitable distance to enable similar determination of the resolution. Nevertheless, as the background/detector noise

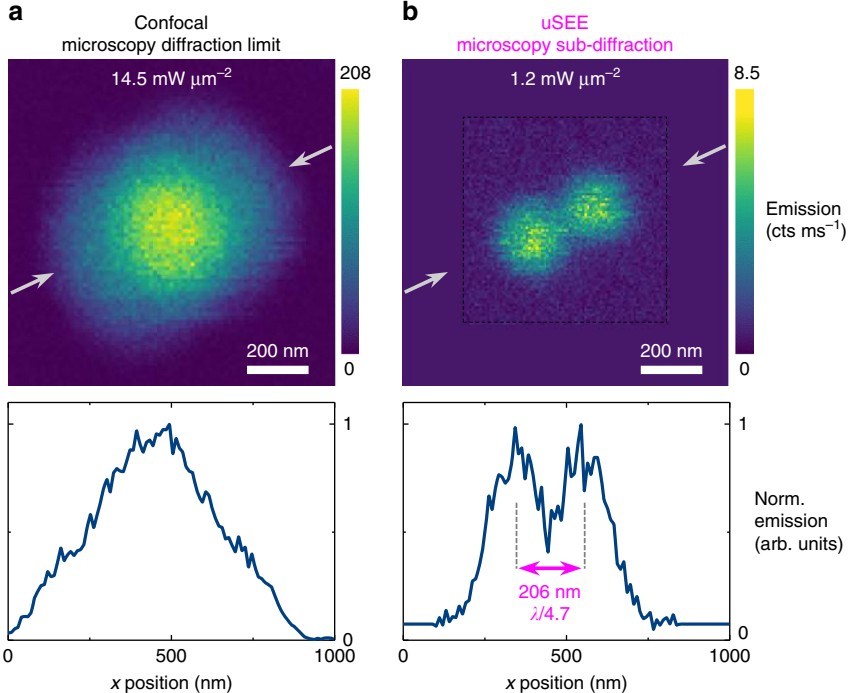

**Fig. 2** Resolving two UCNPs at sub-diffraction distance via uSEE microscopy. **a** Imaging at diffraction-limited resolution does not differentiate the two UCNPs, separated by 206 nm. **b** uSEE microscopy resolves the two UCNPs. The line scans are obtained through a cross section indicated by the gray arrows. The dashed black box indicates the span of the experimentally measured area. The area outside of the black box contains a flat background equal to the average background in the measured area. This facilitates the visual comparison of panels (**a**, **b**)

remains the same, it is safe to assume that FWHM is a good indication for the resolution also in the axial case.

We note that a partially closed pinhole improves the resolution of a confocal microscope[41]. The maximum resolution enhancement factor of $\sqrt{2}$ is reached for an infinitesimally closed pinhole. In our setup, the pinhole is open to at least 1.2 Airy units. This is typically considered as a fully open pinhole (Supplementary Note 2, Section C, F) and a mode of operation that does not lead to resolution enhancement. Thus, the observed resolution improvement in our uSEE microscopy experiments can be solely attributed to the super-linear emission behavior of the nanoparticles.

**Simulation framework**. At present, there is no available theory for predicting the resolution with which a confocal microscope with an arbitrary excitation beam shape images arbitrary non-linear emitters with a continuously changing slope. Such theory is crucial for the successful development and implementation of the SEE microscopy method. In addition, many established and emerging applications often use emitters that, under certain conditions, exhibit a combination of linear, super- and/or sub-linear behavior, and such a theory will benefit these applications too[24,42–46].

The literature typically refers to a factor of $\sqrt{s}$ improvement of the resolution when imaging a non-linear emitter with a slope $s$ in a confocal microscope. However, the simple model predicting this factor, is only adequate along the lateral direction and for a constant-slope emitter, illuminated by a Gaussian beam[29,30]. Thus, the application of this model is extremely restricted, as practically all known emitters have a non-constant slope and, strictly speaking, the assumption of a Gaussian excitation beam is not valid for the high NA objectives typically used in confocal microscopes. The filling factor of the objective and the polarization of the incident light

are also ignored, despite the impact of these factors on the beam shape and eccentricity, which can reach up to $\varepsilon = 0.67$ for high NA objective scenarios[41]. Accounting for the above effects is critical when dealing with non-linear effects, as small deviations in the excitation beam shape and size can result in significant differences in the emission profile, and respectively, the resolution.

We address these challenges and develop a theory that accurately calculates the resolution of a confocal microscope when imaging a non-linear emitter with an arbitrary excitation–emission curve, both for axial and lateral directions.

According to the Rayleigh criteria[40], the resolution is given by the FWHM of the emission profile $I_{em}$ of the particle in the respective direction. The emission profile is entirely determined by the excitation–emission curve $U$ of the particle and the shape of the excitation beam $B_{ex}(\mathbf{r})$ (Supplementary Note 2). In short, we execute the following steps to determine the resolution (for details see Supplementary Note 2, Section A):

1. We calculate the shape of the excitation beam $B_{ex}(\mathbf{r})$, where $\mathbf{r}$ is a position vector, using a full vectorial approach, taking into account the particular components of the setup, the polarization of the incident light and the filling factor of the objective (Supplementary Note 2, Section B). We verify that $B_{ex}(\mathbf{r})$ accurately describes the experimentally observed beam (Supplementary Methods).

2. We measure the emission from a particle for a range of excitation powers, to obtain an empirical excitation–emission curve (data points denoted by the blue dots in Fig. 1b). Fine steps in the excitation power are critical, especially in the highly non-linear, low-power regime. Further details are in Supplementary Note 2, Section D.

3. We interpolate the experimentally obtained points from the excitation–emission curve with a spline function $U$, where

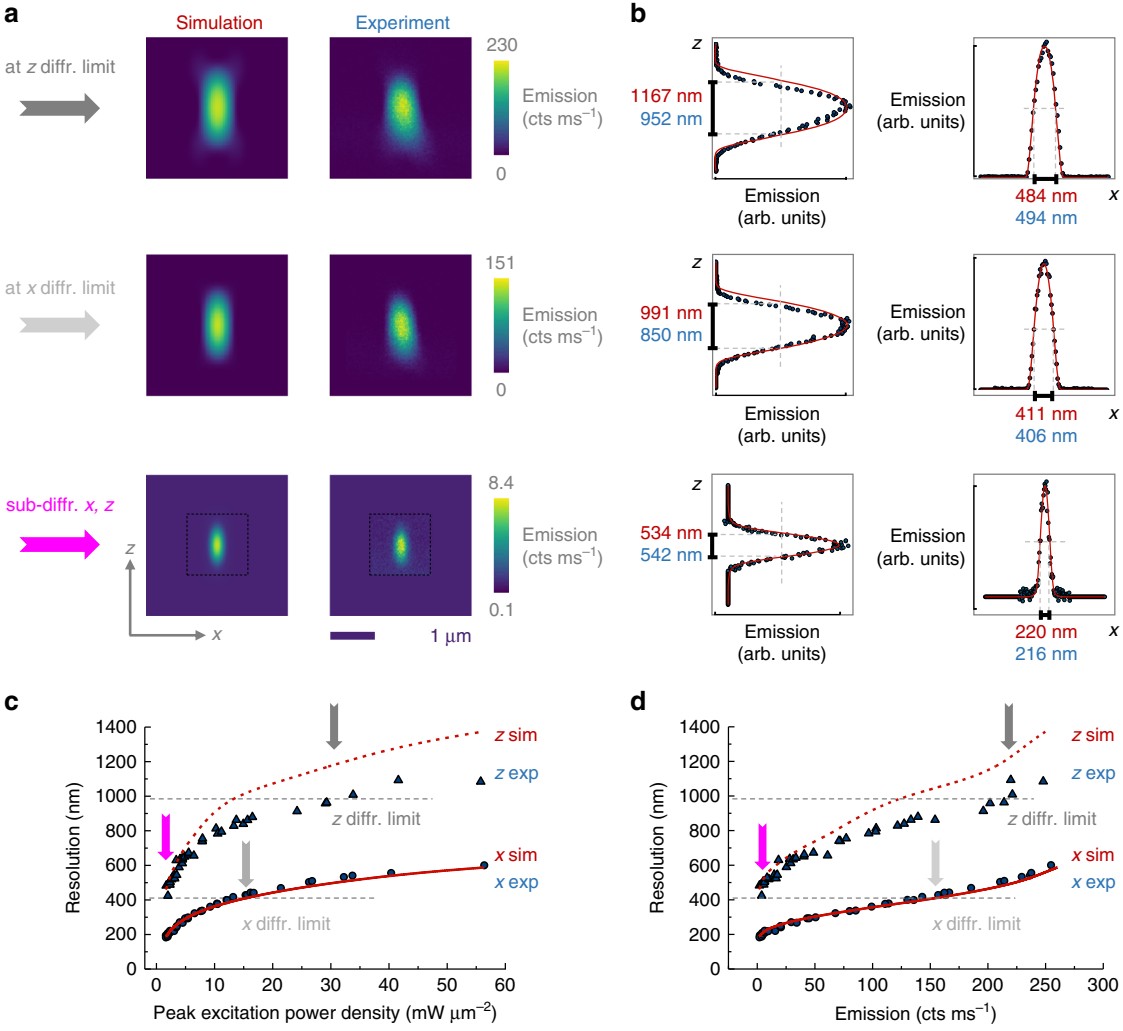

**Fig. 3** The developed theory calculates the 3D SEE microscopy resolution. **a** A good agreement is observed between the simulated (left) and experimental (right) images at three different powers, corresponding to the axial diffraction limit, lateral diffraction limit and sub-diffraction uSEE microscopy. The dashed black box indicates the span of the experimentally measured area. The area outside of the black box contains a flat background equal to the average background in the measured area. This facilitates the visual comparison of the different panels. **b** The cross-sections of the respective profiles in (**a**) quantitatively confirm the agreement between simulation and experiment. Resolution vs excitation (**c**) and resolution vs emission (**d**) curves can be used as a guide to find the most appropriate imaging regime, depending on damage threshold, detection efficiency and desired resolution. Gray dashed lines mark the lateral and axial diffraction limits. The blue symbols represent experimental data (circles for lateral direction and triangles for axial direction). The red curves show the simulated results (full line for lateral direction and dashed line for axial direction)

$I_{em} = U(I_{ex})$. $U$ is plotted as a gray curve in Fig. 1b (Supplementary Note 2, Section E).

4. We use the spline $U$ of the excitation–emission curve and the simulated excitation beam shape $B_{ex}(\mathbf{r})$ to calculate the emission profile $I_{em}(\mathbf{r})$ (Supplementary Note 2, Section C, F):

$$I_{em}(\mathbf{r}) = U(I_{ex}) = U(B_{ex}(\mathbf{r})). \tag{1}$$

The simulated emission profiles are in good agreement with the corresponding experimental profiles, as illustrated for three different excitation power values in Fig. 3a. The chosen excitation intensities correspond to the axial diffraction limit, to the lateral diffraction limit and to an excitation in the low-power SEE microscopy regime. Note, that due to the different profiles of the beam in lateral and axial directions, the diffraction limit is achieved at different powers for the in-plane and out-of-plane directions.

5. We determine the lateral and axial resolution of the system as they are equal to the FWHM of the emission profile in the respective directions (Supplementary Note 2, Section F,

Supplementary Methods). The simulated (red) and the experimental (blue) cross-section profiles are in good agreement (Fig. 3b).

This procedure allows us to plot the FWHM, e.g., the resolution, as a function of the excitation (Fig. 3c) and as a function of the emission (Fig. 3d) across the whole experimentally accessible range of excitation powers. The experimentally measured values are shown with blue symbols (circles for lateral and triangles for axial direction) and the simulation is presented with a red line (full for lateral and dashed for axial direction). The simulated and experimentally obtained values are in good agreement for the lateral direction. For the axial direction, the agreement deteriorates for high FWHM values. We exclude the presence of the pinhole as a cause for this effect, as both experimentally and theoretically we verified that there is a sufficient leeway in the alignment of the pinhole (the detection fiber core): we observe no detectable change of the PSF with displacement of the pinhole within the limits of the open pinhole

condition (Supplementary Note 2, Section C, F). The likely cause for the mismatch is a small tilt of the incident beam (observable in Fig. 3a, right panel) in combination with refractive index variation at the sample interface.

Figure 3c, d indicates that the highest resolution range coincides with the highest super-linear slope supported by the nanoparticle (Fig. 1b). Note that the relation of the slope steepness to the resolution is not explicit in Eq. (1), as the slope is continually changing and remains hidden in the spline of the excitation–emission curve.

The developed theory, and in particular, the graphs in Fig. 3c, d can assist the SEE microscopy user to choose the most appropriate super-linear emitters for their experiment and to adequately tune the experimental conditions, depending on the particular application requirements. An example will be discussed in the next section.

**Comparison of uSEE microscopy performance with different UCNPs.** Irrespective of the specific requirements that different applications of uSEE microscopy might have, commonly, a trade-off between three mutually competing factors is sought-after: (i) high resolution; (ii) low excitation power to avoid damage of biological samples; (iii) strong UCNP emission, which projects into a better signal-to-noise ratio and faster image acquisition times. In this section, we illustrate how the developed theoretical framework can be applied to compare the performance of uSEE microscopy in terms of these features, while employing different emitters.

We consider $NaYF_4$ UCNPs with 20% Yb sensitizer concentration and two different activator concentrations, namely 4% Tm and 8% Tm. Traditionally, UCNPs used for various types of biological applications are doped with relatively low concentration (0.5–1%) of activator ions, as higher concentrations potentially lead to quenching. Recently, several strategies bypassed the quenching effect and showed that high activator doping could increase the brightness and the lifetime multiplexing capabilities of UCNPs in conventional imaging modalities[47,48]. Here, we will investigate the advantages of high activator doping in the particular case of uSEE microscopy.

The uSEE microscopy performance is determined by the excitation–emission curves of the samples (Fig. 4a). The curves are measured on a single particle from the respective sample, identified by STED imaging (Supplementary Methods). We observe a maximum slope of 4 for the 4% Tm doped sample (light green line in Fig. 4a). This is marginally higher than the slopes (3–3.5) currently reported in the literature in the context of sub-diffraction imaging by employing super-linear emitters[29,30]. The 8% Tm doping changes the balance between the different energy re-distribution processes[32], which results in a steeper excitation–emission curve, with a maximum slope of 6.2 (light blue line in Fig. 4a). As the particle emission is stable under such measurement conditions, a pre-illumination procedure has not been applied here and the excitation–emission curve is different from the excitation–emission curve of the pre-illuminated particle used in Figs. 1 and 3.

We input the empirically obtained curves into our computational model and calculate the uSEE microscopy resolution, following the procedure described in the previous section. The resolution is plotted both as a function of the emission intensity (Fig. 4b) and as a function of the excitation power (Fig. 4c). We note that, to simulate the resolution and obtain the full set of graphs, both for lateral (Fig. 4b, c) and axial (Supplementary Note 3) directions, only a measurement of the excitation–emission curve is required. This involves a one-pixel acquisition at each excitation power, which is significantly faster,

compared with obtaining the resolution from experimental PSF images (typically $100 \times 100$ pixels), acquired at each excitation power. Thus, the simulation framework is a powerful tool to rapidly benchmark particle performance in terms of SEE imaging.

The set of graphs presented in Fig. 4 allows us to evaluate the uSEE microscopy performance while employing different emitters. We find the UCNPs with 8% Tm doping a more convenient choice for uSEE microscopy than UCNPs with 4% Tm doping. In the whole experimental range, at a fixed emission value (Fig. 4b) or at a fixed excitation power (Fig. 4c), the 8% Tm doped particles

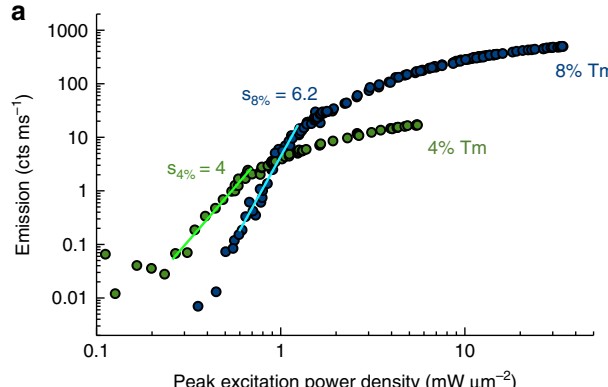

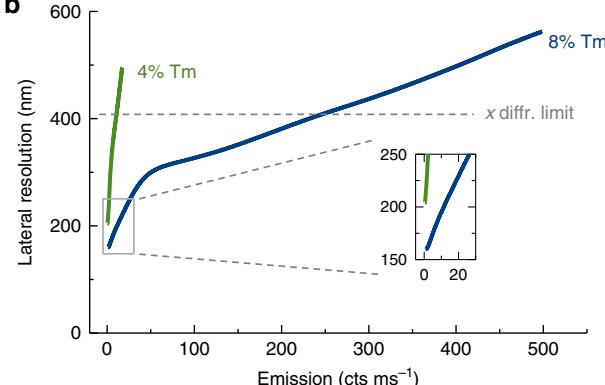

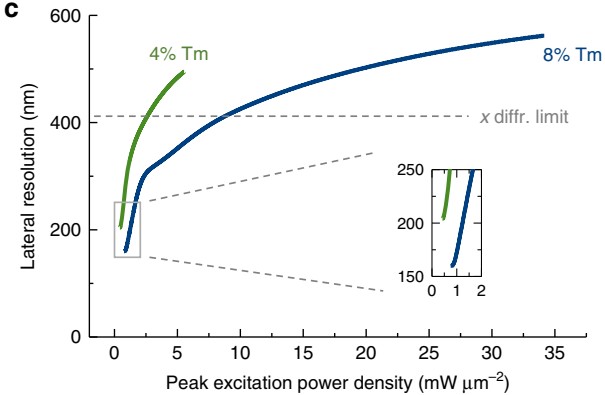

**Fig. 4** High (8%) Tm doping in UCNPs improves uSEE microscopy performance. **a** The 8% Tm doped UCNPs have a steeper super-linear slope than the 4% Tm doped UCNPs. **b** The lateral resolution vs emission graph shows that while imaging at the same resolution, the 8% Tm doped particles have brighter emission. **c** The lateral resolution vs peak excitation power density graph shows that while imaging at the same excitation power, the 8% Tm doped UCNPs offer better lateral resolution. The data for 4% Tm-doped sample are marked in green and the 8% Tm-doped sample in blue

offer better resolution. In addition, for a given resolution value, the higher doped sample exhibits stronger emission (Fig. 4b). This analysis is qualitatively valid for the resolution in axial direction as well (Supplementary Note 3).

For both samples, the best achievable resolution is in the low excitation power regime. Experimentally, the best achievable resolution for the 8% Tm doped sample is 184 nm in lateral and 390 nm in axial direction (Supplementary Note 4). The 4% Tm doped sample is less suited for uSEE microscopy, as the best achievable resolution with these particles is lower – 230 nm in lateral and 607 nm in axial direction. In addition, the imaging of the 8% Tm can be performed more than twice faster at more than twice better signal-to-noise ratio.

The set of graphs in Fig. 4 can also be used as a convenient aid for managing experimental conditions, in response to specific end-user demands. For example, a specific resolution requirement can be satisfied by identifying the corresponding excitation power from Fig. 4c. Alternatively, if a certain UCNP brightness is desired due to restrictions on the detection system, imaging speed or signal-to-noise ratio, Fig. 4b can be used to identify the best achievable resolution by the system under this requirement.

Most of the UCNPs known to date exhibit qualitatively similar behavior in terms of excitation–emission curve shape to the UCNPs discussed here. Often, however, due to the low concentration of activator ions doping, the super-linearity is too weak or occurs at impractically low emission intensity. Thus, the effect of improving the resolution in a confocal imaging setting due to the super-linearity often remains unnoticed. As a rule of thumb, for this type of emitter, we recommend imaging at as low as possible excitation power, as this would lead to better resolution and reduced risk of photo-damaging biological samples. It has to be noted, that this is a unique feature of uSEE microscopy, as other super-resolution techniques need to increase the photon budget in order to reach better resolution[49].

**uSEE microscopy in neuronal cells**. To open the door for biological applications of the uSEE microscopy method, in this section we will verify that the technique can be successfully applied in a biological environment for hours, without compromising the achieved super-resolution and the emission intensity of the nanoparticles.

We functionalized UCNPs (20% Yb, 8% Tm) with colominic acid, an *Escherichia coli* derived substance analog to polysialic acid[50], using a ligand exchange strategy (see Methods). Ligand exchange enabled conversion of the hydrophobic UCNP oleic acid surface to a water-dispersible form and facilitated particle uptake in a cellular model. The functionalized nanoparticles were incubated for 16 h with live neuronal cells differentiated from rat pheochromocytoma cell-line (PC12). Afterward, the cells were washed to remove excess nanoparticles and fixed in formaldehyde. While the function of polysialic acid is still not fully understood, it is known that it plays a role in synaptic formation and plasticity, it influences cell adhesion properties and is involved in the regulation of neuron cell shape, growth, and migration[51].

In order to facilitate navigation in the sample, we added several molecular dyes to the cells (see Methods). Most importantly, we targeted wheat germ agglutinin (WGA)-Alexa Fluor 647 primarily to the cell plasma membrane, which allowed simple verification that the imaged particles were inside cells. The auxiliary laser at 808 nm (functionality discussed in Supplementary Methods) was conveniently sufficient to excite the Alexa Fluor 647 dye, and when coupled with an emission filter centered at 660 nm, allowed observation of the cell plasma membrane.

Several different views of a part of a neuronal cell are presented in Fig. 5: a 3D $z$–stack (Fig. 5a–c) and axial and a lateral cross sections through it (Fig. 5d–f). The (WGA)-Alexa Fluor 647 channel is shown in Fig. 5a, d. The cell membrane and the nuclei are outlined with dashed and dotted white lines, respectively. The dashed gray lines indicate the plane at which we obtained the mutually perpendicular cell cross sections in Fig. 5d–f. Figure 5b, e shows a confocal image of the UCNPs at the lateral diffraction-limited resolution. As discussed in the 'Simulation framework' section, the diffraction limit in lateral and axial direction is occurring at different excitation power. Here, we acquired the images at 11.8 mW μm$^{-2}$ peak excitation power density, which results in imaging with resolution close to the diffraction limit in lateral direction. At this power, the axial resolution is still under the corresponding axial diffraction limit.

A super-resolution image of the same area via uSEE microscopy is obtained simply by reducing the peak laser power density to 1.7 mW μm$^{-2}$ (Fig. 5c, f). We confirmed the achieved 3D resolution by analyzing the zoomed-in areas (Fig. 5g, h), indicated with orange squares in Fig. 5e, f. The resolution achieved by uSEE microscopy in fixed cells is twice better than the diffraction limit in both axial $\lambda/2.2$ (450 nm) and lateral $\lambda/4.7$ (210 nm) directions.

Finally, we verify that the UCNPs can be continuously visualized both in uSEE microscopy conditions and in confocal microscopy at the diffraction limit conditions for periods of more than 5 h in the fixed neuronal cells, without applying a pre-illumination procedure. Under these conditions, negligible deterioration of the emission intensity occurs—below 1% per hour of imaging, while the resolution remains the same (Supplementary Note 1).

## Discussion

In this paper, we make a major step toward enabling 3D super-resolution imaging as an everyday laboratory tool. We demonstrate that 3D sub-diffraction bioimaging can be realized on a conventional confocal microscope, by exploiting the super-linear emission of luminescent markers.

As super-linear emission typically involves multiphoton optical transitions, this approach has sometimes been referred to as multiphoton microscopy. However, this can be misleading, as in the conventional sense of the term, two/three-photon microscopy must double/triple the excitation wavelength. The use of a longer excitation wavelength cancels out any gains in resolution due to super-linearity and as a result multiphoton microscopy is ordinarily not considered as a pathway to super-resolution. In sharp contrast, in the methodology developed here, the luminescent markers allow higher order multiphoton processes to be excited at the same wavelength. In this case, the increase of resolution due to super-linearity allows sub-diffraction imaging. Thus, to avoid terminology confusion, we employ a distinctive name for the method used in this paper, namely we refer to this approach as SEE microscopy if no specific super-linear marker is used, and uSEE microscopy if upconversion particles are employed.

Despite the increasing popularity of UCNPs as biological labels[52], their super-linear properties have often been overlooked in terms of resolution improvement. The most probable reason is the low-doping (0.5–1%) of emitter ions typically used in UCNPs. This often results in negligible resolution improvement or super-resolution occurring at inconveniently low emission intensities. In contrast, here we use NaYF$_4$ nanocrystals with comparatively higher doping concentration (8%) of Tm$^{3+}$ ions. This allows us to reach a super-linear slope of 6.2, which is almost twice steeper than the current literature reports (3–3.5). Moreover, we demonstrate that the highly doped particles reach superior

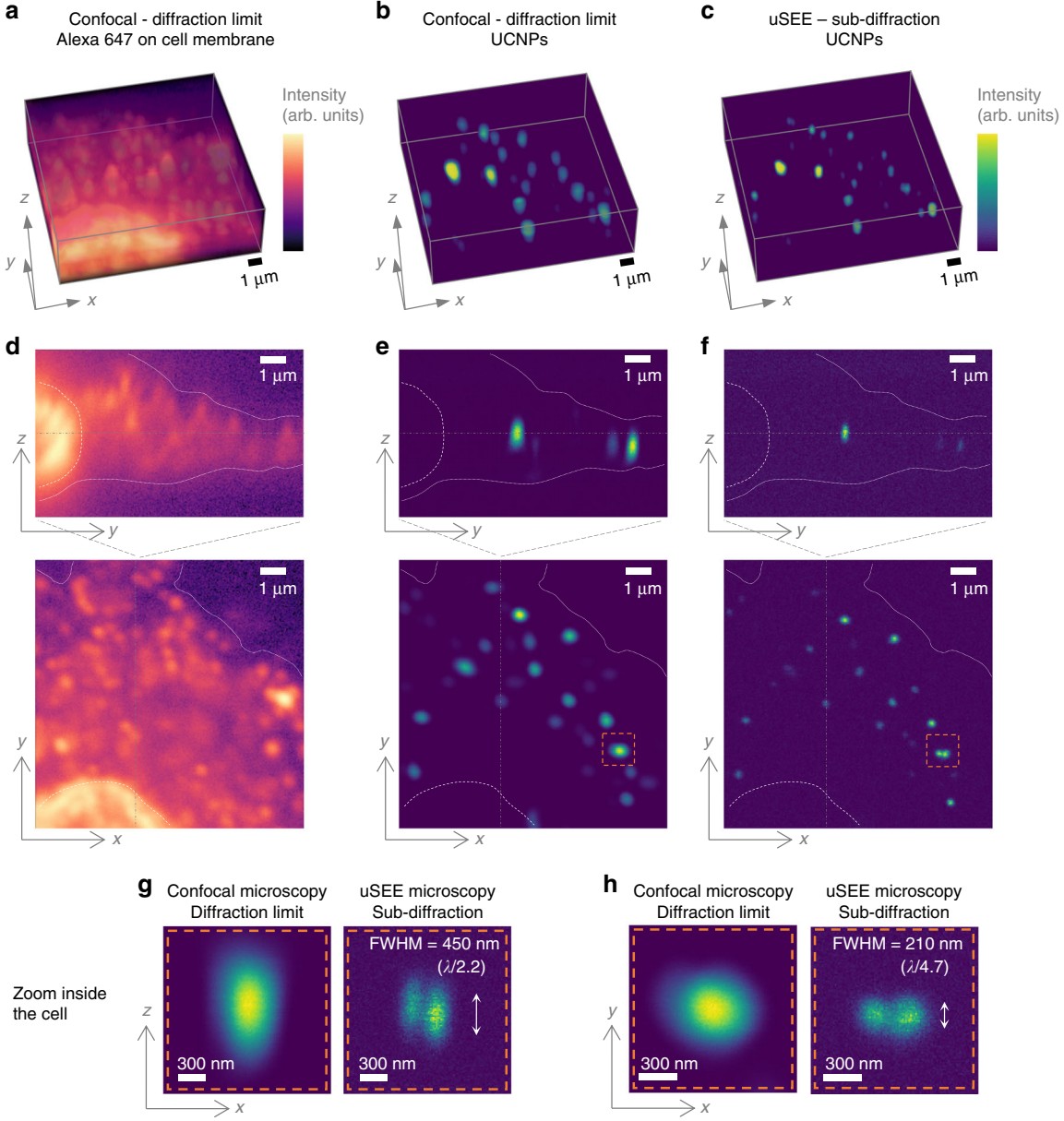

**Fig. 5** 3D sub-diffraction uSEE microscopy imaging in neuronal cells. A confocally imaged 3D $z$-stack of a cell incubated with colominic acid functionalized UCNPs (**a-c**). **a** WGA-Alexa Fluor 647, highlighting the cell plasma membrane. **b** Confocal images of the UCNPs inside the cell taken at excitation power density corresponding to the lateral diffraction limit (11.8 mW μm$^{-2}$). **c** Confocal images of the UCNPs inside the cell taken at low powers (1.7 mW μm$^{-2}$), corresponding to sub-diffraction uSEE microscopy. Corresponding axial and lateral cross sections through the 3D $z$-stacks are shown in (**d-f**). The gray lines indicate the position of the mutually perpendicular cross sections. The dashed white line frames the cell membrane and the dotted white line highlights the nuclei. Closed in zoom on particles inside the cell (orange-dashed volume in (**e**, **f**)) confirms a resolution twice better than the diffraction limit both in axial (**g**) and lateral (**h**) directions

ultimate resolution. We achieve resolution twice better than the diffraction limit both in axial and lateral direction at convenient experimental conditions in terms of excitation power, emission intensity and pixel dwell time. This capability can be easily unlocked for a large number of laboratories already using UCNPs, by simply tuning the imaging conditions (excitation power) or by adjusting the UCNP composition (emitter doping).

In absolute values, the resolution achieved here with uSEE microscopy at near-infrared wavelengths (200 nm lateral, 450 nm axial) is comparable to diffraction-limited resolution at visible wavelengths. Importantly though, uSEE microscopy provides a super-resolution method for the near-infrared range, where such methods are far from abundant. Expanding the currently very

sparse library of probes and super-resolution methods, operating in the near-infrared range, has become an important quest in recent years[14,15]. From one side, working at near-infrared wavelengths has significant benefits in biology, including less scattering/absorption, low photodamage, and low autofluorescence background. From the other side, the long wavelengths in the near-infrared range, intrinsically result in diffraction-limited imaging at more than twice worse resolution, compared with visible light. In reality, the resolution improvement factor achieved with uSEE microscopy ($\lambda$/4.7 lateral, $\lambda$/2.2 axial), is comparable or better than the one achieved with several main-stream super-resolution methods, which require specific setups or image processing, namely SIM, 4Pi, and Airy scan microscopy. Compared with super-resolution techniques employing

UCNPs as luminescent labels in biological setting, uSEE microscopy provides lower lateral resolution (200 nm vs 60–80 nm in refs. [38,53]), but higher axial resolution (450 nm vs 1000 nm in refs. [38,53]). Importantly, uSEE microscopy operates at 2–3 orders of magnitude lower laser power compared with other strategies reported for super-resolution imaging of UCNPs[33,38,53]. This is a significant advantage when imaging biological specimens.

We show that uSEE microscopy super-resolution images can be obtained in fixed neuronal cells for hours, with negligible deterioration of signal intensity and image resolution. UCNPs are becoming increasingly popular as biological markers and numerous strategies for their functionalization and specific labeling of various cellular structures have been proposed[54,55]. Colominic acid, used for functionalization in our paper, and its analogs, sialic and polysialic acid, are non-immunogenic and biodegradable, and therefore, they are attracting interest for drug delivery applications[56]. This opens up promising perspectives toward applications in nanoscale photodynamic therapy and molecular dynamics studies. UCNPs are increasingly popular as drug carriers due to their ability to photorelease drugs with specific timing, exact dosage and high spatial accuracy[55]. It has been shown that development of drugs with organelle specificity can improve drug efficacy[57]. uSEE microscopy can assist this research by enabling long-term observation of UCNP-based drug carriers, both in terms of their pathways and drug release locations, with sub-diffraction spatial accuracy within the targeted organelles. As UCNPs have the ability to enter neuronal cells, another promising bio-application of uSEE microscopy opens up in the field of optogenetics[58]. Here, uSEE microscopy will allow simultaneous sub-diffraction imaging and high-precision optogenetic activation/inhibition of signaling processes in the central nervous system of live animals via cranial window implants[59].

The SEE microscopy concept can be readily implemented in conjunction with other microscopy techniques involving intensity gradient illumination, for example, SIM, STED, light-sheet, Airy scan microscopy, endoscopy, etc. This will improve their performance, especially in terms of axial, lateral resolution and photon budget. For example, non-linear SIM (NL-SIM) typically exploits non-linear fluorophores excited in saturation regime, and as the method requires obtaining tens of images to reconstruct one super-resolved image, its performance is limited by bleaching effects[60]. Moreover, saturating the fluorophores often involves high laser powers, risking damage to biological samples and promoting extensive bleaching. In contrast, if a super-linear regime of the luminescent markers is used (as proposed here with UCNPs), resolution improvement via NL-SIM could be gained at much lower laser fluence and with exceptional photostability.

While UCNPs are a convenient choice for SEE microscopy, neither the SEE microscopy concept nor the developed theoretical framework within this paper, are restricted to this type of luminescent markers. Recent developments in material science bring hope that novel types of super-linear emitters, suitable for SEE microscopy, will emerge in the near future[23,24,43]. Our theoretical framework takes as an input the empirically measured excitation–emission profile of the super-linear emitters and accounts for the concrete confocal microscope used for the measurement. Thus, it can be readily used to assist the development and benchmarking of any type of super-linear emitters. Currently, SEE microscopy yields similar resolution improvement to many popular super-resolution techniques (SIM, 4Pi, Airy scan imaging), while allowing much easier implementation on existing confocal microscopes. Steeper super-linear slope and shorter excitation wavelengths of the luminescent labels would push the achievable resolution of SEE microscopy even further, while shorter luminescence lifetimes, brighter particles[34,35] and/or combination with multi-focal approaches can allow faster imaging[61,62].

## Methods

**UCNP nanoparticle synthesis**. A two-step synthesis method, comprising organometallic growth of core nanocrystals followed by hot-injection homoepitaxy, was employed to produce $NaYF_4$:Yb,Tm UCNPs[33].

First, to obtain the core nanocrystals, 6 ml methanol solution containing 2 mmol of $LnCl_3$ was added together with 12 ml oleic acid and 30 ml 1-octadecene into a 100 ml three-neck round-bottom flask. Ln denotes a mixture with ratios Ln = 76%Y/20%Yb/4%Tm for 4%Tm doped particles and Ln = 72% Y/20%Yb/8%Tm for 8%Tm doped particles. Continuously stirred at 500 rpm under argon flow with volume flow rate of 227 ml/min, the mixture was heated to 30 °C and kept at that temperature for 30 min to remove oxygen. Subsequently, the mixture was heated to 75 °C and kept for additional 30 min to remove methanol. Finally, the mixture was heated to 170 °C forming transparent, pale-yellow solution after 30 min. The solution was then cooled down to 30 °C, and 5 ml of methanol solution containing 4 mmol $NH_4F$ and 2.5 mmol NaOH were added and stirred for 30 min. Afterward, the mixture was slowly heated to 150 °C and kept for 20 min under argon flow with volume flow rate of 227 ml/min to remove methanol and residual water. The argon flow rate was then reduced down to 27 ml/min and the mixture was quickly heated to 300 °C to trigger nanocrystal growth. The mixture was maintained at 300 °C for 1.5 h. Next, the mixture was cooled down to a room temperature, and the synthesized core nanocrystals were isolated by addition of ethanol and subsequent centrifugation. After washing with cyclohexane/ethanol several times, the core nanocrystals were redispersed in cyclohexane at a concentration of 20 mg/ml.

In order to get the precursor solution for hot-injection, we followed the above procedure but instead of heating the mixture to 300 °C, the precursor solution was cooled down to room temperature.

With the core nanocrystal and the precursor solution prepared, we performed the epitaxial growth by adding 3 ml of the core nanocrystal solution to a 100 ml flask together with 10 ml oleic acid and 10 ml 1-octadecene. The mixture was heated to 150 °C and kept for 30 min under argon flow. Then, the mixture was heated to 300 °C and 0.25 ml of the precursor solution was injected into the mixture and incubated at 300 °C for 4 min. This injection and incubation step was repeated 85 times. Then, the mixture was cooled down to a room temperature, and the fully formed nanocrystals were collected according to the same procedure used for the synthesis of core nanocrystals.

**Sample preparation of UCNPs on a coverslip**. Coverslips (Grale HDS, HD LD2222 1.01P0, 22 × 22 mm, No. 1, nominal thickness 0.13–0.17 mm) were washed in pure ethanol, followed by Milli-Q water, partially dried out using nitrogen and finally placed on a filter paper to fully air-dry. A 100 µl drop of Poly-L-lysine solution (0.01% w/v in $H_2O$) was pipetted on the front surface of the cleaned coverslip. The coverslips were subsequently air-dried for 35 min, before being washed three times with 1 ml of Milli-Q water.

The original solution of UCNPs in cyclohexane with concentration of 20 mg/ml was sonicated for 10 min before each use and subsequently diluted to a concentration of 0.05 mg/ml. Next, 20 µl of the diluted solution has been pipetted on the Poly-L-lysine treated coverslip. After 5 s, the coverslip was washed twice by 100 µl of cyclohexane and left to fully air-dry.

Finally, 8 µl of the index-matching medium is pipetted on a cleaned microscopy slide (Thermo Scientific, S41014A, 76 × 26 mm) and the above prepared coverslip with UCNPs is gently placed on top. Slight pressure is applied with tweezers on top of the coverslip to distribute the index-matching medium in the formed chamber. The chamber is then sealed with nail polish, and the sample is left to solidify for at least 48 h.

The index-matching medium is prepared by mixing 6 g of glycerol with 2.4 g of Mowiol 4-88 in a 50 ml centrifuge tube and stirred for 1 h using a magnetic stirrer. Next, 6 ml of water is added and the resulting mixture is stirred for a further 2 h. Afterward, 12 ml of Tris-HCl buffer (0.2 M, pH 8.4) is added and the solution is heated up to 50 °C in a water-bath and subjected to constant agitation until the Mowiol 4-88 completely dissolves. Finally, the mixture is centrifuged at $7500 \times g$ for 30 min to remove any remaining undissolved solids.

**Surface modified UCNPs with colominic acid**. Colominic acid functionalized UCNPs (CA-UCNPs) were produced using ligand exchange: oleic acid capped 8% Tm doped UCNPs at 1 mg/ml w–v concentration in 300 µl of chloroform were added slowly (12 h) into 400 µl of Milli-Q water (pH 7) containing 1 mg/ml w–v concentration of colominic acid in a shaker at 500 rpm. After complete transfer of the UCNPs from the chloroform aqueous phase to water phase, the water phase layer containing CA-UCNPs was removed and centrifuged at 14000 rpm for 20 min. The CA-UCNPs were washed three times in Milli-Q water. The resulting CA-UCNP pellet was redispersed in Milli-Q water and stored at 4 °C until use.

**Cell culture**. Rat PC12 phenotype neuronal cells (ATCC, CRL - 1721) were grown in a Complete Medium which consists of Dulbecco's modified Eagle's medium (DMEM) with high glucose that is supplemented with 10% fetal bovine serum (FBS) (Australian Origin, Life Technologies), 1% antibiotic-antimycotic (10,000 IU/ml penicillin, 10,000 µg/ml streptomycin, and 25 µg/ml of Fungizone R Antimycotic; 15240062 Life Technologies), and additionally supplemented with 5%

Normal Horse Serum. Cells were maintained in a humidified incubator with 95% air and a 5% $CO_2$ atmosphere at 37 °C. All cells were sub-cultured for super-resolution uSEE imaging experiments onto sterilized coverslips (Grale HDS, HD LD2222 1.01P0, 22 × 22 mm, No. 1, nominal thickness 0.13–0.17 mm) for 24–48 h inside 6-well plates in a Complete Medium. PC12 cell's Complete Medium was replaced with nerve growth factor (NGF) containing media for 36 h to encourage neuronal differentiation. The NGF containing media was made using DMEM with high glucose, 1% FBS, 1% antibiotic-antimycotic, 2 mM L-Glutamine, and 100 ng/ml NGF. Finally, 10 µg of CA-UCNPs were added to each well and incubated for a further 16 h in a Complete Medium.

**Fixation and staining of prepared cells for uSEE microscopy**. Culture media and any remaining unbound particles were removed from each well. Coverslips were first washed in 1x PBS pH 7.2, fixed using 4% formaldehyde for 10 min and then washed three times with 1x PBS on a shaker at room temperature. All cells were incubated at room temperature with the F-actin stain Alexa Fluor 594 phalloidin (Molecular Probes, 300 units, A12381) for 10 min at a concentration of 1 unit/ml per well in 1x PBS according to the manufacturer's protocol. Following three additional washes in 1x PBS, cells were incubated with 1 µg of the biotinylated cell plasma membrane marker wheat germ agglutinin (Sigma Aldrich, Biotin-WGA; L5142) per well for 15 min at room temperature. After three washes in 1x PBS, 4 µg of Streptavidin-Alexa Fluor 647 (Molecular Probes, S21374) was added to 1 ml of 1x PBS per well for 1 h at room temperature on a shaker. Finally, the cells were washed three time in 1x PBS and mounted onto slides with Prolong Gold mounting media (refractive index 1.47) containing DAPI (Molecular Probes, P36935).

## Data availability

The data that support the findings of this study are available from the corresponding authors upon reasonable request.

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

## Acknowledgements

This work has been supported by the Australian Research Council (ARC) funding through the Centre of Excellence for Nanoscale BioPhotonics (CE140100003). M.P., L.P., and Y.L. acknowledge support through Discovery Early Career Research Awards (DE170100241, DE180100206 and DE170100821).

## Author contributions

D.D. and M.P. conceived the project, developed the computational model, designed and built the instrumentation, performed the experiments and analyzed the data. M.D. and L.P. developed the UCNP bio-conjugation procedure and prepared the biological samples under the direction of N.P. X.Z. and Y.L. grew the UCNPs and prepared the samples. A.O. helped to conceive the experiments and the theoretical framework, and contributed to analysis and interpretation of the data. J.P. was responsible for the overall direction of the UCNP growth and advanced bioimaging program. D.D. and M.P. wrote the paper with contributions from all authors.

## Additional information

**Competing interests:** The authors declare no competing interests.

