## [Peer Review File · Nature Communications]

Reviewers' Comments:

Reviewer #1:

Remarks to the Author:

The authors described a sub-diffraction imaging method referred as the SEE microscopy by exploiting super-linear emitters. I have the following comments:

It is well-known that the diffraction limit can be beaten in imaging assisted with some nonlinear multiphoton material, because the light intensity is larger than the threshold only at the tip of the focused beam. For a similar reason the high resolution of the SEE microscopy is obvious due to the optical nonlinearity of the upconversion nano-particles (the high order multiphoton process). The four-photon luminescence and the power density dependence curve of Tm³⁺ has also been reported before. It seems that the authors just applied the nonlinear four-photon luminescence in scanning imaging and obtained the nice but expected results.

It is obvious that the resolution degrades as the laser power (exceeding the nonlinear threshold) increases. This has nothing to do with "super-linear" property/region.

Since the non-linear property of upconversion luminescence offers the high resolution and thus the pinhole is not needed. Consequently no need to conduct the experiment with a confocal microscope.

Though the results look nice, the novelty is not sufficient for a high-quality journal like Nature Comm..

Reviewer #2:

Remarks to the Author:

The work by Denkova et al. reports about the usage of upconversion nanoparticles (UCNP) for confocal scanning microscopy. Very interestingly, such UCNP exhibit a super-linear fluorescence response upon excitation, which intrinsically increases the obtainable resolution in confocal fluorescence microscopes. The authors demonstrate a two-fold increase of resolution compared to the diffraction limit. In this case a resolution of ~ 200 nm (Figure 2) is achieved. The UCNP exhibit an exceptionally strong dependence upon excitation intensity: ~4 photons are gained in emission upon excitation with one photon. The manuscript is very well written and easy to follow. The simulations and the demonstration of UCNP in a biological context show the strong effort of the authors, which I respect.

However, my main concern is the maximum achievable resolution of "only" ~200 nm, which is of course directly linked to the long excitation wavelength of 976 nm (which is usually not available in standard confocal microscopes). If one would use standard fluorophores, which can be excited at 488 nm (a standard wavelength for confocal microscopes used in biology) a resolution of ~200 nm can be also obtained with high-numerical aperture objectives. Is there any advantage regarding the long excitation wavelength compared to the short excitation wavelength?

At the end of page 1 the authors state that "...this simple idea has not found practical realization." Maybe the authors are not aware of the following publication, which is not cited: Quantum Dot Triexciton Imaging with Three-Dimensional Subdiffraction Resolution by Hennig et al., Nano Letters, 9, 2466-2470, (2009). In this work by Hennig et al. a resolution improvement of 1.7-fold is demonstrated (also with biological samples) at an excitation wavelength of 445 nm, resulting in a resolution of ~120 nm.

Regarding the simulation framework, the authors claim that there is no available theory for predicting the resolution of confocal microscopes with super-linear emitters. Here I have to disagree, because a similar theory was already developed in ref. 18 of the manuscript.

A minor point:

In the introduction the authors mention tracking and that tracking requires sub-diffraction resolution. That is correct, but irrelevant here, because their method is based on confocal scanning microscopy, which cannot be used for single particle tracking.

In summary, I'm unable to recommend publication of this work in Nature Communication, due to

the lack of novelty. However, the work is very solid and definitely deserves publication, but in a more specialised journal.

Reviewer #3:

Remarks to the Author:

In the paper titled '3D subdiffraction imaging in a conventional confocal configuration by exploiting super-linear emitters' the authors describe the use of a new upconverting nanoparticle that allows them to increase the resolution of a confocal microscope.

The paper is well written. The advantages of the technique that they are proposing in order to increase resolution are numerous, and the major current limitation today is the absence of good probes. Adding to the small repertoire of probes that allow for this type of nonlinear high resolution imaging is of great value and an important contribution to the field and the prospect of the technique becoming one day routine. In light of this, I recommend publication.

Small points to potentially address:

1 - Figure 1a: maybe add the light path trace to show the beam as it collimates and focuses, to stress the point that the technique will not work on a widefield microscope and needs a point focus on the sample.

2 - It is good that the authors considered in depth the effect of the pinhole on the resolution, especially since the simulations don't capture very well the z-resolution improvement. Is it possible and straightforward to check this experimentally? Else, maybe reword the sentence in the abstract 'We develop a computational framework which accurately calculates the 3D resolution improvement across the entire excitation power range'.

3-The sentence "The 8%Tm doped UCNPs have significantly brighter emission at negligible stronger excitation power" could use numbers instead of 'significantly' and 'negligible'. The same for "The brighter emission, combined with a shorter lifetime..."; what is the lifetime?

4-The statements that the particles allow for imaging in a biological context is a bit strong given that the particles were not used to image a biological structure. Maybe offset this in the discussion by covering the cons of the particles and possible solutions, pointing the reader to functionalization protocols and delivery methods that can facilitate labeling.

5- The emphasis on superresolution is objectively true, but maybe misleading. It would be nice to state at some point that if all you were after was resolution, then linear imaging with a short wavelength probe would be comparable to this technique, but that the high-photostability, the long wavelength and the particles compatibility with a standard confocal are all good things that could allow for long term and deep imaging with superresolution given the long wavelength. Maybe even mention long wavelengths in the abstract.

Reply to Reviewer 1:

We are grateful to the Reviewer for evaluating the manuscript and for the positive comments concerning the quality of our results.

We respectfully disagree with the statement related to the lack of novelty in our work. As it can be seen below, none of the Reviewer's comments refers to the main original achievements which we report, namely:

1. First demonstration of the concept in a 3D biological environment using a popular near-infrared biological probe - upconversion nanoparticles
2. Improvement of the resolution in the sub-diffraction regime by lowering the excitation power, opposite to all other super-resolution methods
3. Development of a novel theory, which:
 - A) Calculates the resolution for arbitrary characteristics of the fluorescent probe and setup
 - B) Provides practical guidelines on how to optimize the probes and the imaging conditions

Our arguments supporting the above claims are detailed below as answers to Reviewer #1's comments:

R1.1 "It is well-known that the diffraction limit can be beaten in imaging assisted with some nonlinear multiphoton material, because the light intensity is larger than the threshold only at the tip of the focused beam. For a similar reason the high resolution of the SEE microscopy is obvious due to the optical nonlinearity of the upconversion nano-particles (the high order multiphoton process). The four-photon luminescence and the power density dependence curve of Tm³⁺ has also been reported before. It seems that the authors just applied the nonlinear four-photon luminescence in scanning imaging and obtained the nice but expected results."

This comment re-phrases our introduction, without touching upon the points of the paper where we claim novelty.

Our manuscript states that the concept of improving resolution by using super-linear properties has been suggested "at the end of last century" (page 1, right column, second paragraph). However, Reviewer #1 does not comment on the fact that "some multiphoton nonlinear material", suitable for super-resolution imaging with this approach, was a purely hypothetical material at that time. As we point out in the introduction (page 2, top left paragraph), and Reviewer #3 agrees, the main roadblock towards the realization of the concept has been the lack of suitable probes. We have already explained in the manuscript that the super-linear excitation-emission curve of UCNPs has been reported before (page 2, right column, middle, in the originally submitted version of the paper). Yet, these reports were not exploring the super-linearity in the context of the super-resolution method we realize in our paper.

We respectfully disagree that applying the super-linearity of UCNPs for sub-diffraction imaging is "obvious", because, as explained in the paper, tuning the particle composition and the imaging conditions is a non-trivial task and the community has heavily overlooked the combination of the two concepts. According to Web of Science, out of >4,000 papers published on the upconversion nanoparticle topic in the last 10 years (many published in Nature family journals and cited in our manuscript), no papers demonstrate 3D improvement of resolution due to super-linearity, there are no observations of this effect in a biological sample, and there is no published theoretical framework or practical guidelines how to choose adequate working conditions or how to optimize the probe material in order to achieve super-resolution. This illustrates that our results are far from "obvious" and "expected".

While the applications of UCNP probes for bio-imaging are constantly growing, the super-resolution methods which are available with such probes are currently extremely limited and suffer from the typical problems of the mainstream sub-diffraction microscopy – complex setup, image processing artefacts

and high laser powers (2-3 orders of magnitude higher than the power reported in our paper) (Zhan *et al*, Nat. Commun. **8**, 1058 (2017), Chen *et al*, Nat. Comm **9**, 3290 (2018)). Our work provides unprecedentedly simple solution, which can immediately enable the broad UCNF community to achieve sub-diffraction imaging in their own experimental setting.

R1.2 "It is obvious that the resolution degrades as the laser power (exceeding the nonlinear threshold) increases. This has nothing to do with "super-linear" property/region.

There seems to be a conceptual misunderstanding here: both the super-linear ($s > 1$) and the sub-linear ($s < 1$) regimes are non-linear regimes ($s \neq 1$). As the linear regime is between the super-linear and sub-linear regimes (page 4), it is not clear what the Reviewer means by "nonlinear threshold". The different regimes are defined in the paper as "linear for $s = 1$, super-linear for $s > 1$ and sub-linear for $s < 1$." (page 2, right column, middle in the originally submitted manuscript) and "when we use the term super-linear emitter, we are referring to a non-linear emitter in its super-linear regime." To avoid further confusion, we have now included this clarification in a visual manner in Fig.1b.

To provide a comprehensive overview of all possible conditions the users can encounter when imaging UCNPs, we briefly explain all three regimes in the paper (page 4). However, our main focus is on SEE microscopy, which is defined in the paper as *super-linear* excitation emission microscopy. Thus, it only concerns the super-linear regime and the important and novel claims of the paper are related to this regime. It is not clear why in the context of our work, which aims to improve the resolution while using the super-linear regime, the Reviewer is concerned with a regime which according to his own words "has nothing to do with "super-linear" property/region" and, moreover, degrades the resolution. This comment of Reviewer #1 does not seem to be related to any of the novelty claims of the paper, as it refers to a regime in which SEE microscopy is not defined.

Importantly, we claim a unique feature of SEE microscopy: that *in the super-linear region* where SEE microscopy is defined, the resolution increases by decreasing the excitation power, opposite to all other super-resolution methods. This extraordinary power vs resolution scaling has far-reaching implications for imaging the typically sensitive biological specimens without causing photo-damage. We demonstrate continuous UCNF imaging in fixed neuron cells for more than 5 hours with no significant bleaching and with stable sub-diffraction resolution (Supplementary Note 1). It is essential to distinguish between the resolution improvement by decreasing the power in the saturation and the sub-diffraction regime. While the former is well-known, the latter is a unique feature of our work. We have now emphasized this with an additional sentence in the manuscript (page 2, top right).

R1.3 "Since the non-linear property of upconversion luminescence offers the high resolution and thus the pinhole is not needed. Consequently, no need to conduct the experiment with a confocal microscope."

These points are already stated in our manuscript:

- We are clearly stating that a pinhole is not needed in our case: "*In our setup, the pinhole is open to at least 1.2 Airy units. This is typically considered as a fully open pinhole (Supplementary Note 2, C, F)*" (page 5, last paragraph before "Simulation Framework" Section). It has to be noted that the pinhole cannot be disregarded lightly, without verifying that the pinhole does not play a role in improving the resolution in the particular imaging setting (Reviewer #3 has commented that it is "good" that we consider this effect) - detailed discussion of this non-trivial topic can be found on the first 6 pages of the SI and in Chapters 3, 4 in the book "Principles of Nano-optics" by Prof Lukas Novotny.
- We are clearly stating that SEE microscopy is not restricted to confocal microscopes: "*The SEE microscopy concept can be readily implemented in conjunction with other microscopy techniques (SIM, STED, light-sheet, Airy scan microscopy, endoscopy, etc) for improving their performance, especially in terms of axial, lateral resolution and photon budget.*" (page 10, right column, first

paragraph in the originally submitted manuscript). The experiments are performed on a confocal microscope as this is the most commonly accessible modality in bio-imaging facilities, but it is a unique advantage of our work that both the probes and the theory can be applied in other super-resolution imaging modalities.

We would like to highlight that again, none of these comments are related to or are undermining the novelty claims of the paper.

R1.4 Though the results look nice, the novelty is not sufficient for a high-quality journal like Nature Comm..

We appreciate that Reviewer #1 acknowledges that the results are “nice”, but we kindly disagree with his objection on the novelty of our paper. As illustrated in the replies above, unlike Reviewer #2 and #3, Reviewer #1’s comments are not related to any of the novelty points of our manuscript. The comments are either related to “obvious” points which we have already discussed in the paper (R1.1 and R1.3) or on conceptual misunderstandings (R1.1, R1.2).

Reply to Reviewer #2:

We would like to thank the Reviewer for the insightful comments and the encouraging opinion that our work should be published.

Reviewer #2 raises several points to which we would appreciate a chance to reply, including the novelty aspect with respect to the manuscript which he/she cites – Hennig *et al*, NanoLett 2009 and to Ref 18 (Ref 21 in the revised manuscript). We would like to highlight here again the novelty points of our work:

1. First demonstration of the concept in a 3D biological environment using a popular near-infrared biological probe - upconversion nanoparticles
2. Improvement of the resolution in the sub-diffraction regime by lowering the excitation power, opposite to all other super-resolution methods
3. Development of a novel theory, which:
 - A) Calculates the resolution for arbitrary characteristics of the fluorescent probe and setup
 - B) Provides practical guidelines on how to optimize the probes and the imaging conditions

*R2.1 “At the end of page 1 the authors state that “...this simple idea has not found practical realization.” Maybe the authors are not aware of the following publication, which is not cited: Quantum Dot Triexciton Imaging with Three-Dimensional Subdiffraction Resolution by Hennig *et al.*, Nano Letters, 9, 2466-2470, (2009). In this work by Hennig *et al.* a resolution improvement of 1.7-fold is demonstrated (also with biological samples) at an excitation wavelength of 445 nm, resulting in a resolution of ~120 nm.”*

We thank the Reviewer for bringing our attention to the work using quantum dots as an alternative probe to achieve super-resolution through “multi-photon” process (we have now included a citation of this work in our manuscript – Ref. 25). Despite the obvious merit of the work and the demonstration of the technique in a biological setting, we are convinced that our manuscript has several crucial novelty points and advantages compared to the publication by Hennig *et al*:

- Hennig *et al.* do not present a theory for calculating the increase in resolution. The theory we develop (Novelty #3) provides a crucial advance in the field, as it enables the user with unprecedented flexibility in predicting the resolution while *independently varying both the probe and the setup*. The theory opens a way to benchmark fluorescent labels with completely different photo-physics and evaluate their SEE microscopy performance in different setups and experimental configurations. It allows to easily combine the SEE microscopy method with other imaging methods and evaluate the improvement of their resolution and performance.

[Redacted]

Thus, the developed theory has the potential to be applied in a broad range of imaging applications, from point scanning techniques to light-sheet based approaches, making SEE microscopy attractive to a wide spectrum of audiences. No other paper, including Hennig *et al* allows such flexibility.

- Our work provides first experimental evidence of resolution increase both in axial and lateral directions, simply by *decreasing* the excitation power (Novelty #2). *All* other types of super-resolution techniques require *increasing* of the photon budget in order to improve the resolution. For example, in STED, the depletion beam power has to increase in order to reach better resolutions, PALM/STORM/ SIM techniques require multiple image acquisition steps, with additive illumination costs, to increase the resolution, *etc.* Similarly, in the paper mentioned by Reviewer #2 of Hennig *et al*, the authors use a tri-exciton process to achieve super-resolution, which occurs at one to two orders of magnitude higher excitation power (0.1-1 kW/cm²) than the mono/bi-exciton process yielding a diffraction limited resolution (10-100 W/cm²) – page 2467, right column. Thus, the paper by Hennig *et al* does not undermine Novelty #2.
- As Reviewer #3 points out, the major current limitation of this method is the absence of suitable probes. We recognize that, at visible wavelengths, QDs can be used for such application as described by Hennig *et al*. It has to be noted that a related manuscript (Biochimica et Biophysica Acta **1803**, 1224–1229 (2010)) reveals that the super-linear behavior is not trivially achieved in QDs and “So far, QDot655 are the only commercially available quantum dots that exhibit this phenomenon” (p1228, second paragraph, left in the cited paper). QDs have restricted options for achieving higher order multi-photon processes (Tahara *et al.*, Nat Comm **9**, 3179 (2018)), and therefore, limited scope for further improvement of the achievable resolution. We use a common type of UCNPs for bio-labeling (Novelty #1), and we show a simple pathway for tuning the achievable resolution by changing the doping concentration of the particles. As explained in the reply to the next point, probes working in the near-infrared region, have significant advantages for biological applications and recent publications show a potential to increase the resolution further by an additional order of magnitude.

We have refined the sentence at the end of page 1 which Reviewer #2 finds unsuitable. Additionally, we have clarified the above points in the related parts of the “Introduction” Section to specify the novelty of our work with respect to the paper by Hennig *et al*.

R2.2 “However, my main concern is the maximum achievable resolution of “only” ~200 nm, which is of course directly linked to the long excitation wavelength of 976 nm (which is usually not available in standard confocal microscopes). If one would use standard fluorophores, which can be excited at 488 nm (a standard wavelength for confocal microscopes used in biology) a resolution of ~200 nm can be also obtained with high-numerical aperture objectives. Is there any advantage regarding the long excitation wavelength compared to the short excitation wavelength?”

- Advantages of long wavelengths: Indeed, as Reviewer #2 points out, the main reason for the resolution being “only” 200 nm is the long excitation wavelength of UCNPs. On one side, probes working at visible wavelengths, QDs of Hennig *et al*. included, achieve better resolution simply due to the smaller diffraction limit of light at these wavelengths. On the other side, working at near-infrared wavelengths has significant benefits in biology, including less scattering/absorption, low photodamage and low autofluorescence background. As the diffraction limited resolution at these wavelengths is much worse, expanding the sparse library of probes and super-resolution methods, operating in the near-infrared range has become an important quest in the last years (Lukinavičius, G. *et al*. Nature Chemistry **5**, 132–139 (2013)). UCNPs appear as promising near-infrared bio-

markers and various biological applications have been demonstrated (Hong, G. *et al*, *Nature Biomedical Engineering* **1**, 0010 (2017)). Yet, sub-diffraction techniques using UCNPs “as labels are in its infancy” (Vicidomini *et al.*, *Nat. Meth.* **15**, 173 (2018)). Compared to these methods (Ref. 33 in the revised manuscript: Liu, Y. *et al.* *Nature* **543**, 229 (2017); Ref. 38: Zhan, Q. *et al.* *Nature Communications* **8**, 1058 (2017); Ref. 53: Chen, C. *et al.* *Nature Communications* **9**, 3290 (2018)), uSEE microscopy provides lower lateral resolution (~70 nm in Ref. 33, 38 and 53 vs ~200 nm with uSEE microscopy), but higher axial resolution (~1000 nm in Ref. 33, 38 and 53 vs ~400nm with uSEE microscopy). Most importantly, compared to these methods, uSEE microscopy operates at 2-3 orders of magnitude lower laser power and does not require a purpose-build setup. We have included additional paragraphs in the “Introduction” and in the “Discussion” sections of the revised manuscript to elaborate this comment of Reviewer #2.

- Additional perspectives for improving the resolution: Besides the long wavelength, the other factor determining the resolution is the super-linear behavior of the particles. The particular composition of UCNPs we chose offers super-linear slope twice higher ($s=6$) than other super-linear slopes reported so far in this imaging context ($s=3$). There is a broad scope for further optimization of the super-linearity properties. UCNPs have rich photo-physics with high tunability and we believe bringing the super-linearity as an important parameter on the radar of the UCNP synthesis community will spur the development of probes offering higher resolution improvement. There are already reports of UCNP probes with extremely high super-linear slopes ($s=32$) – Zheng *et al*, *Opt and Laser Technology* **63**, 39 (2014). While the authors have not realized that the observed super-linear effect can be used to improve the imaging resolution, such probes could squeeze the uncertainty imaging volume by a factor of 200 compared to the diffraction limit. The last paragraph of the paper is discussing such perspectives.

R2.3 “Regarding the simulation framework, the authors claim that there is no available theory for predicting the resolution of confocal microscopes with super-linear emitters. Here I have to disagree, because a similar theory was already developed in ref. 18 of the manuscript.”

We respectfully disagree with this comment. In the reply below, we outline the novelty points and advantages of our theoretical framework, compared to Ref. 18 (Ref. 21 in the revised manuscript):

- A unique feature of our theoretical framework is that it allows simulations with an *arbitrary* shape of the excitation beam and estimates the resolution for an *arbitrary* shape of the detection PSF. This feature has important implications, as it opens new research avenues related to combining SEE microscopy with existing microscopy modalities for improving their performance.

[Redacted]

The theory in Ref. 18 does not allow such calculations as it is applicable only for a Gaussian shape of the excitation beam and the detection PSF.

- An important novelty point of our theory is that it can be readily generalized to an *arbitrary excitation-emission profile of any probe*, irrespective of the physics behind the light emission of that particular probe. Such universal theory allowing benchmarking of different probes is crucial for expanding the library of super-linear emitters and popularizing SEE microscopy. The theory in Ref. 18 is applicable only for the particular donor-acceptor type of photo-physics observed for the probes in Ref. 18.
- An original development in our theory is the inclusion of several experimentally sensitive factors, like the polarization effects and the objective filling factor. If omitted, they can lead to miscalculation of the FWHM of the excitation PSF by more than 30% (Ref. 35 in the original manuscript). While stronger super-linearity of the fluorophore leads to higher super-resolution in SEE microscopy, it also results in stronger amplification of this type of errors throughout the optical

system. Thus, it can lead to significant inaccuracy in the theoretical calculation of the expected resolution. Therefore, the inclusion of such experimental factors, allowing accurate calculations, is crucial for the further development of the SEE microscopy method towards achieving higher resolution. Ref. 18 is not accounting for these factors.

In line with the comment of Reviewer #2, we have specified our claim to avoid further confusion about the novelty of our theoretical framework (first sentence from the “Simulation framework” Section). Additionally, we have included a sentence clarifying these points and their implications in the “Introduction” Section. Such a paragraph already exists in the “Discussion” section.

To summarize, we believe the replies above and the clarifications introduced in the revised manuscript will convince Reviewer #2 that our work provides significant advance in the field and opens new research directions, in comparison to Hennig *et al* and Ref. 18, and deserves publication in *Nature Communications*.

Reply to Reviewer #3:

We are grateful to Reviewer #3 for the insightful reading of our paper. We are happy to see that he/she agrees that our work is “of great value and an important contribution to the field” and recommends publication in Nature Communications.

Reviewer #3 raises “small” points which we address below and in the revised manuscript:

R3.1 “Figure 1a: maybe add the light path trace to show the beam as it collimates and focuses, to stress the point that the technique will not work on a widefield microscope and needs a point focus on the sample.”

The light path trace in Fig. 1 a has been adjusted as recommended. Additionally, we have clarified that the schematic explanation in that figure is valid for any scanning type of microscope.

Actually, the benefits of SEE microscopy can be extended even further - to any technique involving illumination with intensity gradient. An example about the advantage the super-linear properties of UCNPs can bring to non-linear SIM is included in the “Discussion” Section.

R3.2 “It is good that the authors considered in depth the effect of the pinhole on the resolution, especially since the simulations don’t capture very well the z-resolution improvement. Is it possible and straightforward to check this experimentally? Else, maybe reword the sentence in the abstract ‘We develop a computational framework which accurately calculates the 3D resolution improvement across the entire excitation power range’.”

On one side, we agree with the Reviewer that this sentence can be misleading, and we have reworded it in the revised version of the manuscript.

On the other side, we have verified both experimentally and by simulations that the reason for such mismatch is not in the position of the pinhole, as the Reviewer assumes. There is a sufficient leeway in the alignment of the pinhole (detection fibre core), with no detectable change of the PSF observed with displacement of the pinhole within the limits of the open pinhole condition (Supplementary Note 2, C, F). To avoid further misunderstandings, we have now included an additional sentence clarifying this in the Simulation Framework Section (page 7, left, bottom of the third paragraph) and in Supplementary Note 2, F.

Instead, we believe that the observed difference is due to a small experimental tilt of the beam (visible in Fig. 3a, page 6, right), and due to imperfect refractive index matching at the interface (such asymmetry

can be observed in Fig. 3b). To help direct the reader towards this evidence in the manuscript, we have now referred the reader to the concrete panel illustrating this in Figure 3.

As the observed mismatch is attributed to inevitable, small experimental misalignment, we believe that it is not undermining the validity of our theoretical framework.

R3.3 “The sentence “The 8%Tm doped UCNPs have significantly brighter emission at negligible stronger excitation power” could use numbers instead of ‘significantly’ and ‘negligible’. The same for “The brighter emission, combined with a shorter lifetime...”; what is the lifetime?”

We agree with the Reviewer that concrete numbers are always a more solid indicator in such statements. In qualitative aspect, the trend stated in these sentences can be observed in Fig. 4. However, the specific values change by orders of magnitudes across the excitation power range and quantifying the trend in a sentence becomes impossible. Thus, we believe that the best course of action in this case is to remove the generalizing sentences which the Reviewer points out and to cite the relevant numbers only in the lowest power regime, where SEE microscopy is most powerful and most likely to be used. These changes are implemented in the revised manuscript.

R3.4 “The statements that the particles allow for imaging in a biological context is a bit strong given that the particles were not used to image a biological structure. Maybe offset this in the discussion by covering the cons of the particles and possible solutions, pointing the reader to functionalization protocols and delivery methods that can facilitate labeling.”

We agree with the Reviewer that, although our UCNPs are functionalized with colominic acid, which, in principle, allows further functionalization with antibodies and respectively, specific binding in cells, in our case we have not labelled a concrete biological structure. In-line with the request of the Reviewer, in the revised manuscript, we have re-worded the phrase “biological context”. Indeed, numerous procedures for functionalization of UCNPs and labelling of biological structures already exist, together with particular biological applications. Following the Reviewer’s recommendation, we have now offset this discussion to the “Introduction” and “Discussion” Sections and amended the requested citations.

It is worth pointing out that some applications, for example in optogenetics context, do not necessarily require specific labelling of the UCNPs – Chen *et al* in Science, 2015 (Ref. 58) have achieved optogenetic activation of the brain of a living mouse while using non-functionalized UCNPs, directly injected in the brain of the animal. SEE microscopy would allow not only imaging, but also activation of these particles with higher, sub-diffraction precision. Thus, we believe that the importance of our work for the biological community is not undermined by the particular choice of a labelling procedure or biological structure.

R3.5 “The emphasis on superresolution is objectively true, but maybe misleading. It would be nice to state at some point that if all you were after was resolution, then linear imaging with a short wavelength probe would be comparable to this technique, but that the high-photostability, the long wavelength and the particles compatibility with a standard confocal are all good things that could allow for long term and deep imaging with superresolution given the long wavelength. Maybe even mention long wavelengths in the abstract.”

We thank the Reviewer for this suggestion and we have implemented it by several sentences in the “Discussion” section, in the “Introduction” Section and mentioning the long wavelength in the abstract.

To summarize, in the revised manuscript we have addressed all minor points for corrections which Reviewer #3 has recommended before publication in *Nature Communications*.

Reviewers' Comments:

Reviewer #1:

Remarks to the Author:

The authors admit that the concept of improving resolution by using super-linear properties and the super-linear excitation-emission curve of UCNPs has been reported before. Indeed, the four-photon process of Nd³⁺-UCNPs has been discovered and the visible-to-visible four-photon ultrahigh/sub-diffraction resolution microscopic imaging under the excitation of 730-nm diode laser has been realized, and the resolution has reached 161 nm (Wang et al, Optics express, 24.2 (2015), A302-A311). Sun's group developed the high-resolution multiphoton microscopy with a low-power continuous wave laser pump, and the axial and lateral resolutions were improved approximately 1.5 times compared with confocal microscopy. (Chen et al, Optics letters, 43.4 (2018): 699-702). In summary, although the authors' work is well organized and presented, the novelty and the improvement of the so-called SEE microscopy is limited.

Reviewer #2:

Remarks to the Author:

I believe that the manuscript by Denkova et al. is now in much better shape. The authors addressed my main concerns regarding the novelty and the manuscript addresses now these points explicitly, especially the benefits of using long wavelengths and low excitation powers. I also have to agree that the theory is more elaborated and provides new guidelines for the development of new probes for SEE microscopy. While I still believe that the individual points made in this manuscript do not provide justification for publication in Nature Communications, I feel that by taking everything together this work should be published in this journal.

Reviewer #3:

Remarks to the Author:

The authors have addressed all suggestions, so I recommend publication.

Reply to Reviewer 1:

We thank the reviewer for the careful assessment of our manuscript, and we are pleased that he/she finds the results “well organized and presented”.

We respectfully disagree with the reviewer’s statement about the “limited” novelty of our work.

We would like to stress again the original achievements of our work:

1. First demonstration of the concept in a 3D biological environment using a popular near-infrared biological probe - upconversion nanoparticles.
2. Improvement of the resolution in the sub-diffraction regime by lowering the excitation power, opposite to all other super-resolution methods.
3. Development of a novel theory, which:
 - A) Calculates the resolution for arbitrary characteristics of the fluorescent probe and setup;
 - B) Provides practical guidelines on how to optimize the probes and the imaging conditions.

Both papers with which Reviewer #1 compares our work have been cited in our manuscript – references 20 and 24 in the original submission and references 24, 30 in the first revision of the manuscript.

We would like to highlight that none of our novelty points are demonstrated in the above (or any other) references. The above references do not provide a demonstration of the technique in a biological environment and in the biologically desirable near-infrared window (Novelty #1), they do not comment on the unique power vs resolution trend which we report in our work (Novelty #2) and they do not provide a universal theoretical platform for the method (Novelty #3).

Below, we will elaborate on these claims and clarify why the Novelty points which we report provide a significant advance in the field and how they open new research avenues in the areas of chemistry, biology and physics.

R1.1 “The authors admit that the concept of improving resolution by using super-linear properties and the super-linear excitation-emission curve of UCNPs has been reported before. Indeed, the four-photon process of Nd³⁺-UCNPs has been discovered and the visible-to-visible four-photon ultrahigh/sub-diffraction resolution microscopic imaging under the excitation of 730-nm diode laser has been realized, and the resolution has reached 161 nm (Wang et al, Optics express, 24.2 (2015), A302-A311).”

- Our work realises 3D sub-diffraction imaging in a biological environment with near-infrared probes (Novelty #1). No such claim is made by the manuscript of Wang *et al*, which, as Reviewer #1 points out, uses ‘visible-to-visible’ probes, in a non-biological setting. On one side, probes working at visible wavelengths, like the ones chosen by Wang *et al.*, allow better resolution simply due to the smaller diffraction limit at these wavelengths. On the other side, imaging at near-infrared (NIR) wavelengths offers significant benefits in biology, including low photodamage, less scattering/absorption and low autofluorescence background. The diffraction limited resolution at the NIR wavelengths is much worse, thus expanding the sparse library of probes and super-resolution methods, operating in the NIR range has become an important quest in the last years (Lukinavičius, G. *et al. Nature Chemistry* **5**, 132–139 (2013)). We use excitation wavelength which is significantly longer (980 nm) than the one used by Wang *et al.* (730 nm), yet our lateral resolution (180 nm) is similar to the resolution observed by Wang *et al.* (160 nm). Importantly, we also report an improvement of the resolution compared to the diffraction limit in axial direction (more than 2 times). No super-resolution in the axial direction

is shown in the paper of Wang *et al.* In summary on Novelty #1, we believe that our demonstration of the method beyond the visible wavelengths into the near-infrared region, combined with the first-time 3D application of the method in a biological environment opens the door for a broad interest in this technique from the field of biology – for example, for direct application in the areas of optogenetics and sub-cellular drug-delivery targeting, as elaborated in the Discussion section of the Manuscript.

- Our work provides the first realization, both experimentally and theoretically, of resolution improvement by *decreasing* the excitation power (Novelty #2). This unconventional trend is extremely beneficial for biological applications where photon budget management is of key importance. It is worth to note that all other super-resolution methods require increasing of the photon budget to improve the resolution – STED requires increasing of the depletion beam power, PALM/ STORM/ SIM – like techniques require more image acquisitions. The unconventional trend which we report as Novelty #2 has not been discussed by Wang *et al.*
- Finally, we present a comprehensive theory that can be applied to an *arbitrary excitation-emission profile* of any probe, irrespective of the physics behind the light emission process (Novelty #3). This capability is of crucial importance, as it allows expansion of the currently extremely limited library of suitable super-linear emitters for SEE microscopy. This capability provides the field of synthetic chemistry, for the first time, with a universal tool to evaluate and compare newly developed probes in terms of SEE microscopy performance. Additionally, our theory allows predicting the resolution in both lateral and axial directions, while illuminating with an *arbitrary excitation beam* shape. This capability opens an exciting avenue for combining SEE microscopy with numerous popular imaging modalities, including STED, Bessel Beam imaging, Airy scan imaging, *etc.*, as elaborated in the Discussion section of the manuscript [Redacted]. These unique capabilities are unlocked for the first time by our theoretical framework. No such theory, and respectively, such capabilities, are reported in the paper of Wang *et al.*

As a side note, we would like to make the following comment: the reviewer states that “*the four-photon process of Nd³⁺-UCNPs has been discovered*” and related to a four-photon transition. This statement is correct, however irrelevant, as our paper does not involve Nd³⁺ transitions, neither do we attribute a four-photon process as the emission principle of our particles. As we describe in the manuscript (page 3, bottom right), there are various transitions in the Tm³⁺ ions which are playing a role in the multi-photon emission of our UCNPs, attributed to a combination of 4, 5, 6 or higher multi-photon processes. This allows us to achieve a significantly stronger super-linearity (slope $s > 6.2$) compared to the slopes reported by Wang *et al.* (slope $s < 3.5$).

In order to clarify further the unique novelty points of our paper compared to Wang *et al.*, we have revised the relevant parts of the introduction section (color-coded).

R1.2 “Sun’s group developed the high-resolution multiphoton microscopy with a low-power continuous wave laser pump, and the axial and lateral resolutions were improved approximately 1.5 times compared with confocal microscopy. (Chen et al, Optics letters, 43.4 (2018): 699-702).

Similar to the manuscript by Wang *et al.*, the work of Chen *et al.* employs visible-to-visible probes in a non-biological setting and does not report a universal theory for the method. The paper of Chen *et al.* does not claim any of our novelty points and hence, the bullet point replies to R1.1 also directly address the comparison of our manuscript to the paper by Chen *et al.*

Additionally, we would like to highlight that accessing the super-linear process in nanodiamonds, as presented by Chen *et al.*, requires a sophisticated combination of three different lasers with a certain illumination pulse sequence. In comparison, we access the super-linear regime with a single laser, simply by decreasing the laser power (Novelty #2).

We kindly note that despite the claim of Reviewer #1 for axial resolution improvement in the paper by Chen *et al.*, the best axially achieved resolution is 500 nm (page 702, left column, end of the first paragraph). This value is almost identical to the diffraction limit, which in axial direction is roughly equivalent to the excitation wavelength used by Chen *et al.* - 532 nm and 589 nm. In contrast, we demonstrate resolution improvement in axial direction down to 450 nm, which is more than twice better than the axial diffraction limit at our excitation wavelength of 980 nm.

To highlight further our novelty points compared to the manuscript of Chen *et al.*, we have re-phrased the relevant parts of the Introduction section (color-coded).

R1.3 "In summary, although the authors' work is well organized and presented, the novelty and the improvement of the so-called SEE microscopy is limited."

As justified by the replies above, our work provides unique advantages compared to the papers referenced by Reviewer #1, and generally, to the current literature on the topic. We enable unique capabilities, related to the much-needed expansion of the library of super-linear probes and super-resolution methods in the near-infrared range. Our theoretical framework allows extension of the SEE microscopy method towards novel imaging modalities and combination with other popular techniques. We demonstrate, for the first time, that the method is applicable for 3D imaging in a biological environment and suggest direct implementation towards the fields of opto-genetics and sub-cellular drug delivery.

To summarize, we believe that the replies above and the clarifications introduced in this second revision of our manuscript, will convince Reviewer #1 that the comprehensive methodology package presented in our work, provides significant advances in the field and opens new research directions, in comparison to the papers by Wang *et al.* and Chen *et al.*, and deserves publication in *Nature Communications*.

Reply to Reviewer #2:

We would like to thank the reviewer for the insightful comments and constructive contribution towards improving our manuscript. We are pleased that the reviewer acknowledges the value of the combined methodology which we have developed and recommends our manuscript for publication in Nature Communications.

Reply to Reviewer #3:

We are grateful to the reviewer for the revision of our paper and his valuable comments which helped us to improve our manuscript. We are pleased that the reviewer recommends our manuscript for publication in Nature Communications.